# Study on the Influence of Coal Structure and Oxidation Performance by Endogenous Bacterium

**Xuanmeng Dong** [1,*], **Fusheng Wang** [1,2], **Liwen Guo** [1,2] **and Tiesheng Han** [3,4,*]

1   College of Mining Engineering, North China University of Science and Technology, Tangshan 063210, China; fswang@ncst.edu.cn (F.W.); guoliwen@ncst.edu.cn (L.G.)
2   Mining Development and Safety Technology Key Lab of Hebei Province, Tangshan 063210, China
3   School of Public Health, North China University of Science and Technology, Tangshan 063210, China
4   Hebei Province Key Laboratory of Occupational Health and Safety for Coal Industry, Tangshan 063210, China
*   Correspondence: dongxuanmeng@stu.ncst.edu.cn (X.D.); ts_han@ncst.edu.cn (T.H.)

**Abstract:** In order to solve the defects of traditional coal spontaneous combustion prevention technology in a closed goaf, a strain of aerobic endogenous bacteria was isolated from coal and used as a blocking raw material. Based on the metabolic and reproductive characteristics of microorganisms, the experimental study on the inhibition of coal spontaneous combustion by microorganisms was carried out. The colonies were isolated and purified by the dilution concentration plate method and the scribing plate method. The growth morphology of microorganisms was analyzed, and the growth curve was determined. The strains were identified by seamless cloning technology for high-throughput sequencing. The surface morphology of coal was analyzed by SEM, the differences of oxidation characteristic temperature points were analyzed by TG–DTG–DSC images, a programmed heating experiment was used to analyze the concentration of the indicator gas CO, and the changes in microscopic groups before and after microbial action were analyzed by FTIR and XPS spectra. Therefore, the inhibition of coal oxidation by endogenous bacteria was verified from macroscopic and microscopic perspectives. The results show that the coal bacteria isolated from the coal is *Lysinibacilus* sp. After the culture of *Lysinibacilus* sp., the surface of the coal demonstrated less detritus, and was relatively smooth. In the early stage of low temperature oxidation of coal spontaneous combustion, the characteristic temperature point of coal oxidation and the reaction between coal and $O_2$ could be delayed by *Lysinibacilus* sp., and the total heat release was reduced in the combustion process. Not only that, *Lysinibacilus* sp. could also reduce the CO concentration during coal heating. After the coal was decomposed by *Lysinibacilus* sp., the C=C thick ring skeleton structure had little effect; however, the aromatic substitution pattern changed. This bacterium had an effect on the C-O bond, reducing the percentage of $-CH_2-$ and increasing the percentage of $-CH_3$. It might also use the crystalline water in coal for life activities. The carboxyl carbon in coal changed the most, with a decrease of 12.03%, so it might become the carbon source required for microbial growth. The reproductive metabolism of microorganisms also affected the form of nitrogen, and the percentage of pyridine nitrogen in coal was reduced. The ratio of single-bond carbon to double-bond carbon in raw coal was about 3:2, but after this bacterial action, the ratio of the two was about 1:1. The analytical conclusions of XPS and FTIR spectra were consistent, and the results supported each other.

**Keywords:** microorganism; coal spontaneous combustion; coal structure; characteristic temperature; low-temperature oxidation

## 1. Introduction

In recent years, Chinese coal mining has gradually extended to the deep, which makes the ore pressure appear obviously. Roof subsidence, and drum and surface cracks of closed walls often occur around the closed surface of the goaf. Therefore, the closed goaf has special features, such as not being airtight, multiple and difficult to distinguish wind source

points, and a wide range of air flow. Once a spontaneous combustion is found in it, the ignition point is difficult to determine, and may be continuously burned for long periods. If the discovery is not timely, the delay in solving the fire situation is not conducive to fire prevention work and security [1]. To effectively solve the scientific problem of coal spontaneous combustion in closed goaf, the judgment indicators of ignition degree in closed regions are proposed [2]. The method of predicting the temperature field of spontaneous combustion of residual coal by monitoring air leakage in the goaf was proposed by Zhang et al. [3] Not only is the prediction of closed fire zone studied by researchers, but also the technique of preventing coal spontaneous combustion. Tang et al. [4] found that injecting nitrogen, carbon dioxide, and other gases in the closed fire area of the goaf can reasonably control coal spontaneous combustion to a certain extent. However, the thermal effect can cause more cracks and holes to be formed on the coal surface, and the reaction rate of coal and oxygen is accelerated, in the oxygen-poor state or inert gas environment. Ding [5] analyzed the process defects and shortcomings of liquid nitrogen direct injection type and vaporization type in goaf, so as to improve the inconvenience of poor insulation, easy vaporization and volatilization, low temperature, and pipeline freezing and cracking during long-distance liquid nitrogen transportation. Zhang [6] used low-temperature liquid nitrogen, liquid carbon dioxide injection, glue injection for cooling, and low-temperature plugging and other comprehensive treatment methods to avoid the hidden danger of spontaneous combustion of floating coal in goaf. A completely similar goaf platform device was built by Lu et al. [7] to study the foam fire extinguishing system. Jiang et al. [8] interrupted the chemical reaction of coal oxidation by using compound inhibitors. Environmental protection inhibitors were used by Zhang et al. [9] to inhibit coal spontaneous combustion, which can make the surface of the coal form a liquid film to isolate oxygen, evaporation, and heat.

In fact, the cracks in the closed goaf are interlocking, and the ideal completely closed space cannot be formed, so the injection of inert gas cannot achieve a durable inert. The liquid retarder technology has disadvantages such as instability and short validity period, and the gel organic matter fire extinguishing also has defects such as high cost and easy drying and cracking. Therefore, on the basis of traditional fire prevention, the propagation characteristics of aerobic microorganisms isolated from coal mines are used to put forward the method of microbial inhibition of coal spontaneous combustion. Kang et al. [10,11] cultured four kinds of bacteria to degrade lignite, which reduced the number of N=O and C=O characteristic groups. At the same time, they used another four kinds of bacteria to dissolve lignite, and found that the dissolution rate of *Pseudomonas fluorescens* to coal could reach 61.9%. Microorganisms can destroy loose molecular structures, active functional groups, and compounds containing unsaturated bonds in coal, resulting in structural changes [12]. At the same time, minerals also play an important role in the oxidation and spontaneous combustion of coal [13]. Huggins et al. found that $H_2SO_4$ is one of the byproducts in the process of pyrite oxidation, and coal with higher pyrite content is easier to be oxidized than coal lacking pyrite [14]. M.H. Kiani et al. [15], Cardona et al. [16], and Wang Rui et al. [17] used the mixture of *Acidithiobacillus ferrooxidans* and *Acidithiobacillus thiooxidans* to remove sulfur elements from high-sulfur coal. Therefore, microorganisms can change the microstructure of coal by desulphurization, and the oxidative heat release of coal is reduced, so that spontaneous combustion of coal is inhibited.

On the basis of previous research results, a kind of aerobic coal bacteria was isolated from coal, and morphological observation, growth characteristics, and molecular biological identification were carried out. It was mixed with lignite of different particle sizes to produce retarded coal samples. The programmed heating experiment was used to analyze the concentration changes of the indicator gas CO. Thermal synchronous analysis technology was used to analyze the oxidation characteristic temperature points, and the oxidation characteristics of raw coal and bacterial coal were compared to explore the inhibition ability of microorganisms from a macro perspective. FTIR and XPS were used to compare the transformation of coal structure under the action of microorganisms, and the change law

of coal characteristic groups was revealed from the microscopic perspective. It provides experimental support for the study of microbiological control methods of coal spontaneous combustion in closed goaf.

## 2. Experimental Design and Method

### 2.1. Culture, Identification, and Morphological Characteristics of Coal Endogenous Bacterium

2.1.1. Isolation of Coal Endogenous Bacterium

The bacterium in the coal used in the experiment came from the lignite of Hongliulin coal mine in Shaanxi province to ensure the bacterium is non-toxic and harmless to humans. Sterilized tools are used by professionals to remove surface coal that has been exposed to air for a long time. The inner coal was placed in a 500 mL sterile sampling bottle, encapsulated, and then separated, screened, and purified by a mixture of microorganisms. In an ultra-clean table (ZHJH-C1112C vertical flow type C single working face ultra-clean table in ZhJH-C1112C, Shanghai, China), 10 g of pulverized coal was weighed and placed in a triangle flask containing 100 mL 0.9% NaCl solution. About 15 glass beads were placed in the triangle flask and shaken for 25 min. Through the collision between the glass beads, the bacterium in the coal can be fully dispersed and washed off, reducing the error caused by the uneven mixing of microorganisms in the normal saline. A total of 2 mL of the prepared microbial diluent was added into 200 mL of Luria–Bertani (LB) liquid medium and cultured at 150 rpm in a 30 °C thermostatic oscillator (Shanghai Zhicheng ZWY-2102C vertical double-layer small capacity full temperature oscillator) for 48 h to obtain the enriched mixed bacterial solution [18]. The above bacterial mixture was diluted at $10^{-1}$, $10^{-2}$, $10^{-3}$, $10^{-4}$, and $10^{-5}$, and 10 μL was taken from diluents of different concentrations and coated in LB solid medium, then placed in a constant temperature box at 30 °C for growth, and the growth state was recorded every 12 h. After the isolation and growth of a large area of colonies, a colony with rapid growth was found, temporarily named X. X colonies growing in a single circle were picked out and marked in a plate in the new medium and placed again in the incubator for growth. When colonies of uniform size and morphology were observed under the microscope with consistent Gram staining, the single colony was oscillated in LB liquid medium. We then used 2–5 mL turbid bacterial solution 12,000 g, centrifuged it for 1 min, suspended it with 20% glycerin, and stored it in the freezer at −80 °C for use. The components of medium in the paper are shown in Table 1.

**Table 1.** Main components and functions of culture medium.

| Name | Main Component | Function |
|---|---|---|
| Luria–Bertani (LB) liquid medium | Tryptone, yeast extract, NaCl, deionized water | It enables intensive microbial growth |
| Luria–Bertani (LB) solid medium | Tryptone, yeast extract, NaCl, agar powder, deionized water | It purifies and enriches the target strain |
| Luria–Bertani (LB) solid–coal medium | 200 mesh or more coal powder, tryptone, yeast extract, NaCl, agar powder, deionized water | It verifies whether coal affects the growth of the target strain |

2.1.2. Growth Morphology and Curve of Coal Endogenous Bacterium

A small amount of bacterial suspension was dipped into the inoculation needle and strewed into the LB solid medium. The components of the medium are shown in Table 1. After 48 h, it was observed that the surface shape of the colony was translucent and small and round with smooth yellow edges, as shown in Figure 1a, and the growth ability of bacterium X in the coal environment is shown in Figure 1b. When coal was used as the main component of the medium, the normal growth of bacterium X was not affected. The Gram stain showed blue and purple, belonging to Gram-positive bacteria, being aerobic and rod-shaped. The pictures of bacterium X with different magnification taken by optical microscope (Jiangnan Yongxin BM2000 biological microscope) are shown in Figure 2a,b.

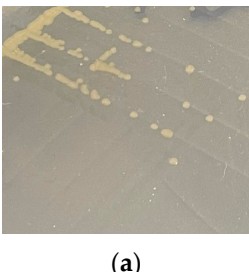
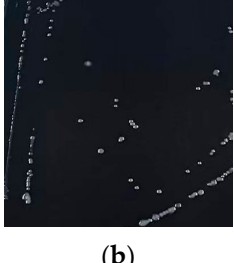

(**a**) (**b**)

**Figure 1.** Morphology of bacterium X in different culture medium. (**a**) LB solid medium; (**b**) LB solid–coal medium.

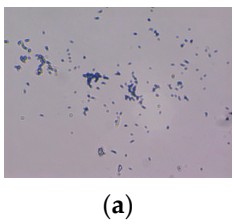
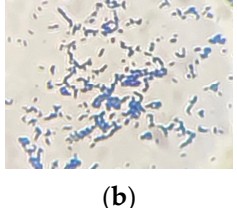

(**a**) (**b**)

**Figure 2.** Gram staining and morphology of bacterium X under different magnification microscopes. (**a**) For $100\times$ magnification; (**b**) for $400\times$ magnification.

Understanding the growth cycle of bacterium X helps to better determine the survival time in the coal environment. A single full colony was selected from solid medium and transferred to liquid medium. The turbidimetric method was used to determine the growth curve of bacterium X. The bacterial suspension concentration is proportional to the light absorption value. The measured light absorption value (OD600) at 600 nm wavelength was taken as the ordinate, and the growth time was taken as the horizontal coordinate. The resulting curve is the growth curve period of bacterium X, as shown in Figure 3. The growth stage is divided into four stages: slow stage, logarithmic growth stage, stable stage, and decline stage [19]. The logarithmic growth stage of bacterium X is about 30 h, and the stable stage is about 18 h.

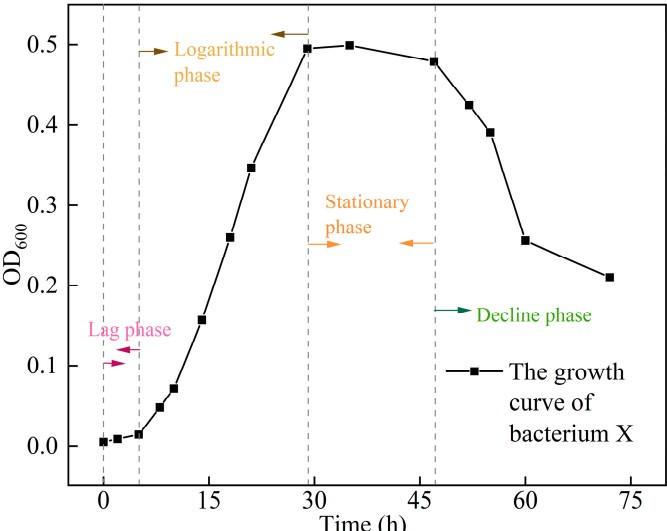

**Figure 3.** The growth curve of bacterium X.

### 2.1.3. Molecular Biological Identification of the Bacterium

In bacterial phylogenetic studies, 16S rDNA is a common and effective molecular clock. It is characterized by few species and high content (accounting for about 80% of

the total bacterial DNA), with a medium molecular weight of about 1.5 Kb (1500 bp), and occurs in all organisms. It can not only express the differences among different bacteria, but also obtain their sequences relatively easily by means of sequencing, so it is recognized by researchers. The molecular biological design and identification operations of bacterium X are as follows [20].

①  Bacterium X was cultured to the logarithmic phase of growth and then collected by centrifugation, and the bacterium was used as a PCR template to design amplification primers for:

F-primer f27: 5′-AGA GTT TGA TCC TGG CTC AG-3′;

R-primer r1942: 5′-GGT TAC CTT GTT ACG ACT T-3′.

Polymerase chain reaction (PCR) amplification DNA reaction system of 50 μL was designed [21]: 19 μL-ddH$_2$O; 25 μL-2×Taq PCR MasterMix II enzyme; 2 μL each of 16S diluted pre- and post-primers; bacteria solution template −2 μL;

②  PCR reaction products were amplified by 1% agarose gel electrophoresis to form bands, and amplified bands were cut and recovered using the OMEGA E.Z.N.A.®® Gel Extraction Kit. Purified genomic DNA of bacterium X was approximately 1500 bp (developed by Azure c500 laser NIR imaging system, USA, as in Figure 4);

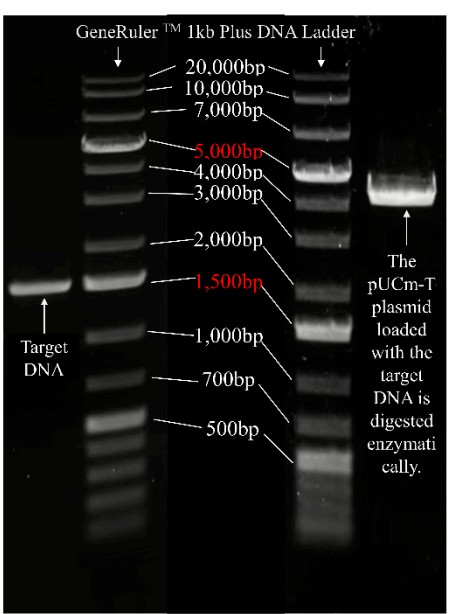

**Figure 4.** The 16S rDNA sequence of bacterium X and the pUCm-T plasmid containing the target sequence were enzymatically cleaved.

③  Purified DNA was ligated in pUCm-T vector and the ligation product was calcium transferred into receptor cells;

④  Blue–white spot screening containing the target gene was performed by X-Gal and IPTG solution, white spot enrichment culture was taken in ampicillin (Amp)-resistant environment, pUCm-T plasmid containing the target gene was extracted and enzymatically cleaved, and then the target gene insertion was successfully verified (Figure 4) and sent to Bioengineering (Shanghai) Co. for sequencing [22];

⑤  DNA sequences were analyzed by BLAST comparison, genus information of bacterium X was obtained, similar DNA sequences were compared, phylogenetic tree was constructed by MEGA11 software (Figure 5), specific taxonomic status and information of closely related species were obtained, and finally bacterium X was detected as *Lysinibacilus* sp.

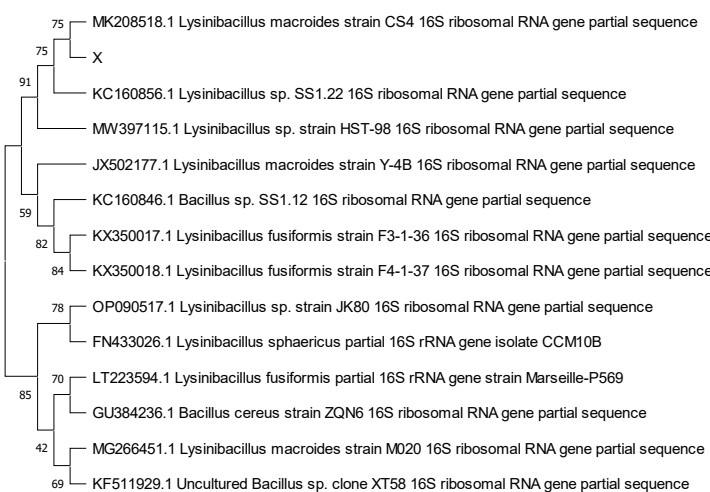

**Figure 5.** Phylogenetic tree of bacterium X.

## 2.2. Coal Sample Preparation

Lignite coal from Hongliulin coal mine was selected as the untreated raw coal sample, identified here as raw coal (RC). To ensure the representativeness of the coal samples, fresh coal samples were taken at every 2 m and the easy oxidized parts of the coal surface were stripped in the coal mining face. The coal cores were taken for crushing and grinding, and the coal grains were sieved to obtain samples with mixed grain size of 0.178 mm to 0.850 mm. The samples to be tested were dried at room temperature for 24 h and stored in sealed bags for spare.

The Hongliulin lignite after the action of *Lysinibacilus* sp. was a bacterially blocked coal sample, identified here as bacterial coal (BC). The 50 g of raw coal sample with mixed particle size was sterilized by autoclaving at 121 °C for 15 min. It was mixed with 200 mL of bacterial solution and placed in a 500 mL sterile conical flask and incubated in a constant temperature shaker (New Brunswick Excella E25 constant temperature shaker, UK) at 28 °C 140 rpm for 14 d. Finally, the coal–bacteria mixture was removed and filtered, washed with deionized water, repeated 3 times and dried at room temperature for 48 h.

The scanning electron microscope (SEM) equipment is Japanese electronic scanning electron microscope JSM-IT100, and the sample structure was observed using secondary electron imaging mode (SE). Figure 6a indicates the untreated raw coal, and Figure 6b indicates the bacterial coal sample. the surface of RC is embedded with granular crystals, a large number of debris distribution, with a scattered and complex shape, polygonal, angular, and strip-like, etc. The surface shows irregular concave pores and folded fissures. The amount of BC surface debris is relatively small, a few stomata, and dissolution pores appear in some areas. The surface is relatively smooth and flat, and linear fissures exist [23]. Table 2 shows the analysis of the main elements of RC and BC. Ultimate analysis was carried out with the Germany Elementar-UNICUBE, with a sample weight of about 0.20 mg. We burned the sample at high temperature until it decomposed, and the elements to be measured were converted into the corresponding gases for analysis. After being acted on by bacteria, the percentages of carbon, hydrogen, and total sulfur of lignite decreased, and the percentages of oxygen and nitrogen increased.

**Table 2.** Ultimate analysis of RC and BC.

| Coal Sample | $C_{daf}$ (%) | $H_{daf}$ (%) | $O_{daf}$ (%) | $N_{daf}$ (%) | $S_{t,d}$ (%) [1] |
|:---:|:---:|:---:|:---:|:---:|:---:|
| Raw coal (RC) | 76.58 | 4.60 | 14.51 | 1.00 | 0.30 |
| Bacterial coal (BC) | 71.82 | 4.41 | 15.42 | 1.37 | 0.28 |

[1] Total sulfur on dry basis.

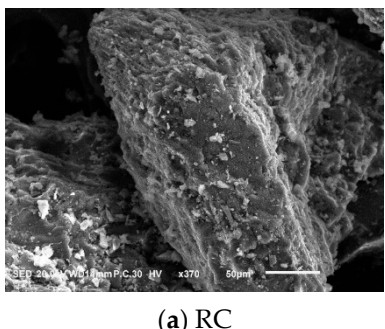 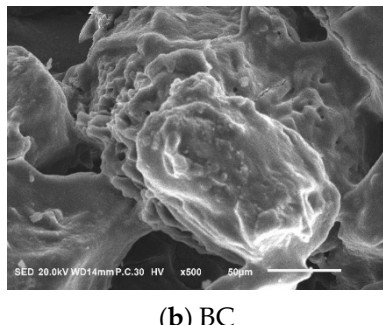

(**a**) RC                        (**b**) BC

**Figure 6.** Scanning electron micrographs of secondary electron (SE) imaging modes of RC and BC.

*2.3. Oxidizing Properties of Coal after Bacterial Reaction*

2.3.1. Programmed Heating Experiment

The dried mixed particle size coal sample of 80 g was put into the tank, the temperature probe was connected to the middle of the tank, the dial showed the temperature of the coal, the initial temperature was set to 25 °C, the set rate of temperature increase was 1 °C/min, and the rate of air intake was 150 mL/min. The gas was collected in the temperature range of 30~250 °C and was passed into the gas chromatograph to detect the concentration of CO. Figure 7 represents a schematic diagram of a programmed heating gas chromatography equipment.

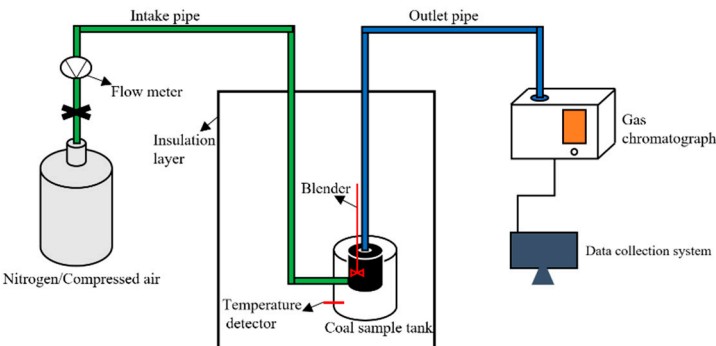

**Figure 7.** Schematic diagram of programmed heating gas chromatography equipment.

2.3.2. Thermal Simultaneous Analysis Equipment and Parameter

The experimental apparatus was Beijing Hengjiu microcomputer differential thermal balance HCT-3, and the coal sample was weighed, and was about 10 mg into a $5 \times 4$ mm size 99% alumina crucible. The gas injected into the simultaneous thermal analysis was compressed air ($V_{oxygen}$:$V_{nitrogen}$ = 1:4) with a flow rate of 10 mL/min, an initial temperature of 25 °C, an end temperature of 820 °C, with the temperature rising by 10 °C per minute, and a sampling period of 10 s/time. The DTG curve was calculated by taking the first-order derivative of the TG curve with respect to the temperature, which indicated the rate of coal mass change during the temperature change.

*2.4. Analysis of Microscopic Group Changes in Bacterial Action*

2.4.1. Fourier Transform Infrared Spectroscopy (FTIR)

The Fourier transform infrared spectroscopy (FTIR) equipment was a VERTEX 80 V infrared spectrometer manufactured by BEUKER, Germany. About 100 mg of potassium bromide was first taken and mixed with 1 mg of coal and ground finely, then the gray powder was placed into the grinding tool and the press was pressed under vacuum. The test background was scanned before the experiment, and the sample was scanned 16 times. As shown in Figure 8, the FTIR curves were mainly divided into 2 large regions, the fingerprint region (600–1300 cm$^{-1}$) and the functional group region (1300–4000 cm$^{-1}$). The

characteristic peaks in them can distinguish the subtle differences of organic structures and are called the fingerprint region, and the functional group region can mainly identify the types of functional groups in organic matter [24].

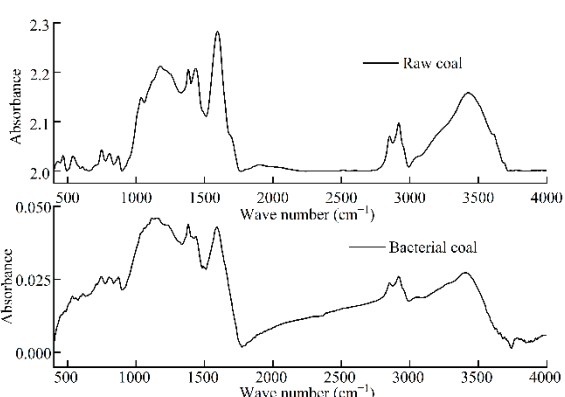

**Figure 8.** Fourier transform infrared spectrograms of RC and BC.

### 2.4.2. X-ray Photoelectron Spectroscopy (XPS)

The coal sample was scanned by X-rays, which ionized the valence electrons of atoms or groups, and the electrons were excited as photoelectrons by photons. The photoelectron energy is used as the horizontal axis and the pulse signal intensity as the vertical axis to derive the photoelectron energy spectrum. Different irradiation intensities corresponded to different electron binding energies, from which the composition of the substance to be measured can be known. The coal samples were dried under vacuum at room temperature for 24 h. The coal samples were analyzed by X-ray photoelectron spectroscopy (XPS) scanning on a Thermo Scientific$^{TM}$ K-Alpha$^{TM+}$ spectrometer using an alpha X-ray source (1486.6 eV) with a power of 100 w under vacuum ($p < 10^{-8}$ mbar). The full=scan transmission energy was 150 eV and the high-resolution scan was 25 eV. Combined with an energy correction standard for the C1s peak (284.8 eV), the experimental peaks were split and peak intensities were fitted by Avantage software, as shown in Figure 9.

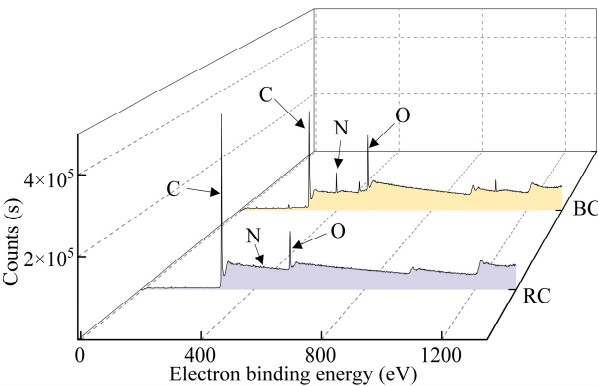

**Figure 9.** XPS spectra of RC and BC.

## 3. Results and Discussion

### 3.1. Coal Spontaneous Combustion Index Gas CO

CO is used as a common indicator gas in underground coal mines, so it is necessary to compare the CO concentration of RC and BC in the pre-spontaneous combustion ignition stage. It can be clearly seen from Figure 10 that the CO concentration increases gradually with the increase in temperature. During the initial oxidation stage (60~80 °C), there is no significant difference in the CO concentration generated by heating between BC and RC. However, during the slow self-heating stage (80~140 °C), the CO concentration generated

by coal after bacterial reaction differs from that of raw coal, with the CO concentration of BC being slightly lower than that of RC. The critical acceleration stage (140~180 °C) is the stage with the largest difference in CO concentration, with that of BC being significantly smaller than that of the RC at the same temperature. During the pyrolysis fission stage (>180 °C), the CO concentration of BC remains consistently less than RC. CO was used as an indicator gas to determine the degree of coal oxidation and it was found that the coal treated with *Lysinibacillus* sp. had a diminished ability to participate in reactions with oxygen.

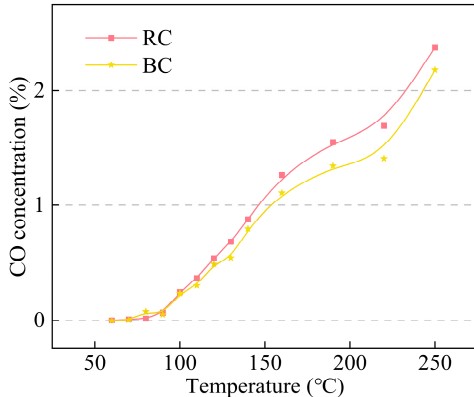

**Figure 10.** CO concentration released by RC and BC during heating.

### 3.2. TG–DTG Curve Analysis

The TG–DTG curves of the two coal samples were used to derive the characteristic temperature points, masses, and their first-order derivative values for RC and BC according to the previous definitions of the characteristic temperature points for the time period from the onset to the combustion temperature, as shown in Table 3 [25]. the TG–DTG curves of RC and BC are shown in Figure 11. $T_0$ is the ambient temperature of 28 °C. $T_1$ is the maximum rate of weight loss after the evaporation of free water and water detachment from the coal during the temperature increase point; the oxidation reaction of coal accelerates the reaction with moisture detachment. $T_2$ indicates the temperature at the minimum value of weight before combustion on the TG curve, called the dry cracking temperature, which indicates the junction point of the moisture desorption coal phase. $T_3$ is the starting point of oxygen absorption weight gain for coal mass balance. $T_4$ indicates the dynamic equilibrium stage, the point where the DTG curve has the maximum rate of weight gain per unit time, and the rate of weight gain after this point starts to decrease. $T_5$ is the temperature corresponding to the maximum value on TG, which is the starting point of the period of thermal decomposition. $T_6$ is the ignition temperature for the vertical line of the main peak of the DTG curve, and its vertical line intersects the tangent of the point on the TG line, and the intersection of the horizontal line of $T_5$, not on the TG curve, which is the value corresponding to the horizontal axis of this intersection, is the ignition temperature. $T_7$ indicates the maximum value of DTG, representing the fastest burning rate per unit time, after which the burning rate decreases. $T_8$ is the temperature at the end of combustion when the mass is stable, called the burnout temperature.

**Table 3.** TG–DTG characteristic temperature points and mass changes of RC and BC at 28~800 °C.

| Coal Sample | | $T_1$ | $T_2$ | $T_4$ | $T_5$ | $T_6$ | $T_7$ | $T_8$ |
|---|---|---|---|---|---|---|---|---|
| Raw coal (RC) | Temperature (°C) | 73.91 | 111.12 | 193.31 | 216.04 | 386.81 | 496.63 | 596.04 |
| | Mass (%) | 98.28 | 96.54 | 95.19 | 94.89 | 71.47 | 25.44 | 2.85 |
| | DTG (%·min$^{-1}$) | −0.0562 | −0.0399 | −0.0064 | −0.0253 | −0.1694 | −0.7299 | −0.0004 |
| Bacterial coal (BC) | Temperature (°C) | 84.22 | 132.14 | 208.78 | 295.51 | 400.23 | 483.93 | 624.55 |
| | Mass (%) | 97.40 | 95.09 | 94.58 | 93.19 | 81.86 | 38.85 | 5.37 |
| | DTG (%·min$^{-1}$) | −0.0735 | −0.0243 | −0.0113 | −0.0361 | −0.2532 | −0.5787 | −0.0152 |

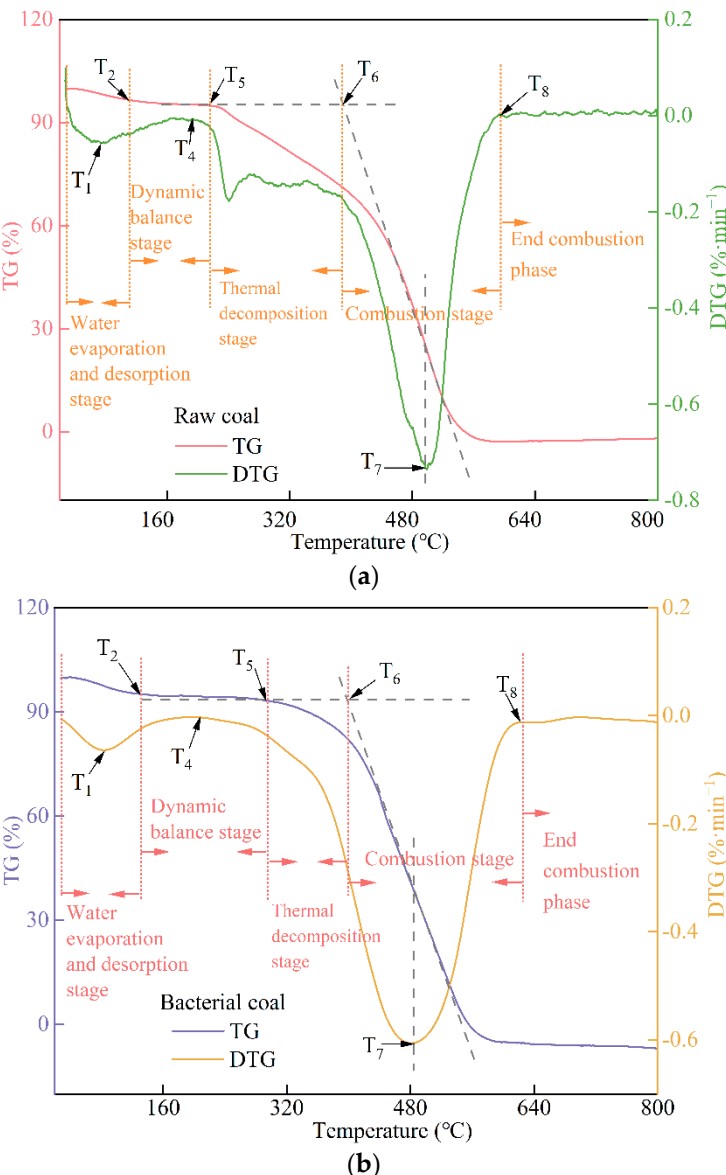

**Figure 11.** TG–DTG curves of (**a**) RC and (**b**) BC.

In the water evaporation desorption stage ($T_0$ to $T_2$), $T_1$ and $T_2$ of BC increase by 10.31 °C and 21.02 °C, respectively, compared with those of RC, indicating that the bacterium could delay the desorption of free water from coal and hinder the precipitation of small molecule gases such as $O_2$, $CO_2$, and $N_2$ physically adsorbed by coal in this stage. In the dynamic equilibrium stage ($T_2$ to $T_5$), which includes the oxygen absorption stage, the temperature point of $T_3$ cannot be determined because the coal is lignite with a low degree of metamorphism, the structure is easily destroyed during the gradual warming process, the small molecule gases are very easily generated through oxidation reactions, and the degree of weight loss is always greater than the degree of oxygen absorption weight gain by chemisorption. $T_4$ and $T_5$ of BC are 15.47 °C and 79.47 °C higher than those of RC, and this bacterium retards the oxidation of coal before the thermal decomposition stage. In the combustion stage from $T_6$ to $T_8$, $T_7$ of BC is smaller than that of RC, and $T_6$ and $T_8$ of BC are larger than that of RC. The conclusion of the maximum combustion rate T7 after the coal has been burned is in agreement with Hong et al. The maximum combustion rate of the bacterial residue corresponds to a significantly lower temperature than that of the coal [26]. This bacterium could not prevent the combustion of coal already in the combustion stage, but before the occurrence of coal spontaneous combustion, in the

low-temperature oxidation stage of coal, *Lysinibacilus* sp. could delay the coal spontaneous combustion to some extent and retard the reaction of coal with $O_2$.

### 3.3. Differential Scanning Calorimetry (DSC) Curve Analysis

Differential scanning calorimetry (DSC) indicates the additional heat required to keep the temperature difference between the experimental material to be measured and the reference material to be measured at zero per unit time within the temperature control to reflect the enthalpy change of the product to be measured versus temperature. Its peak area is the total heat of the reaction at that stage, and the peak shape upward indicates the exothermic reaction. The integration curve of DSC curve indicates the relationship between the heat absorption (release) and temperature in that temperature range, so the temperature point corresponding to the maximum peak of DSC indicates the maximum heat release rate temperature [27]. The DSC curves of RC and BC, and their corresponding integration curves from 28 to 800 °C, are shown in Figure 12, and their characteristic temperature points are shown in Table 4.

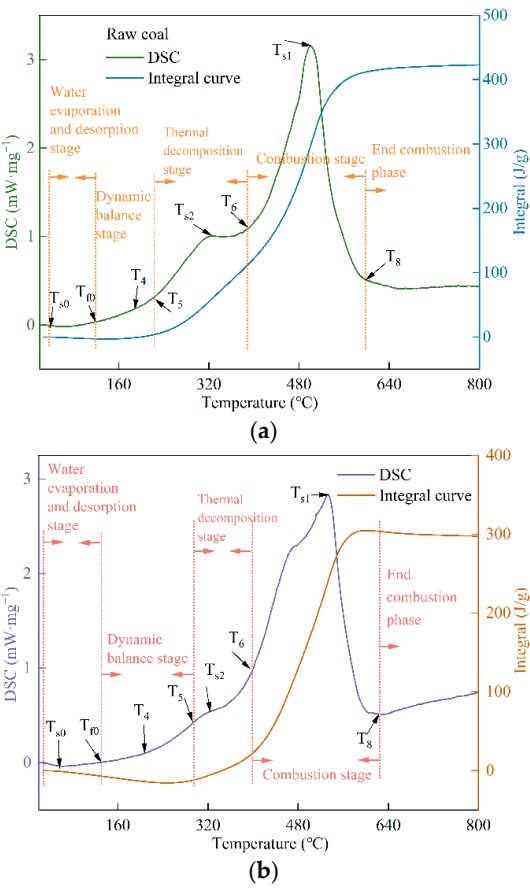

**Figure 12.** DSC curves and corresponding integral curves of (**a**) RC and (**b**) BC at 28~800 °C.

In the water evaporation desorption stage, $T_{S0}$ indicates the maximum heat absorption temperature, and BC is delayed by 20.86 °C compared with RC. $T_{f0}$ indicates the end temperature of heat absorption by water evaporation, which is close to the $T_2$ temperature point on the TG curve. RC is 117.88 °C and BC is 129.83 °C, with a difference of 11.95 °C. At the combustion stage, $T_{S1}$ indicates the maximum peak temperature of DSC in the exothermic phase, which has all higher temperature points than $T_7$. This is due to the increase in temperature caused by heating, which reduces the combustible components of the coal and continually decomposes to produce a large number of gaseous products, thereby driving the enthalpy to its peak. In addition, $T_{S1}$ of BC is 32.02 °C higher than the temperature point of RC. $T_{S2}$ indicates the shoulder peak of the maximum enthalpy peak

temperature point, where BC > RC. The peak area of the integration curve of RC from 28 to 800 °C is 422.82 J/g, and that of BC is 297.69 J/g, indicating that the exothermic heat of the coal subjected to the blocking reaction of *Lysinibacilus* sp. is less than that of the original coal during the warming process from 28 °C to 800 °C. *Lysinibacilus* sp. is not only able to delay the temperature at which the maximum enthalpy of combustion occurs, but also reduces the total heat released from the coal during combustion.

**Table 4.** DSC characteristic temperatures and related parameters of RC and BC at 28~800 °C.

| Coal Sample | | $T_{S0}$ | $T_{f0}$ | $T_4$ | $T_5$ | $T_6$ | $T_{S1}$ | $T_{S2}$ | $T_8$ |
|---|---|---|---|---|---|---|---|---|---|
| Raw coal (RC) | Temperature (°C) | 36.12 | 117.88 | 193.31 | 216.04 | 386.81 | 500.88 | 323.52 | 596.05 |
| | DSC (mW·mg$^{-1}$) | −0.0076 | 0.0318 | 0.1950 | 0.2854 | 1.0796 | 3.1551 | 1.0006 | 0.5177 |
| | Integral (J/g) | −0.05 | −2.83 | −0.56 | 2.58 | 110.18 | 298.74 | 57.85 | 411.40 |
| Bacterial coal (BC) | Temperature (°C) | 56.98 | 129.83 | 208.78 | 295.51 | 400.23 | 532.90 | 324.34 | 624.55 |
| | DSC (mW·mg$^{-1}$) | −0.0328 | 0.0044 | 0.1032 | 0.4323 | 0.9961 | 2.8360 | 0.5426 | 0.5156 |
| | Integral (J/g) | −0.90 | −6.84 | −13.91 | −11.74 | 22.75 | 239.81 | −5.32 | 303.61 |

*3.4. FTIR Fitting Analysis*

The FTIR curves are divided into four sections, and the RC and BC sections correspond to Figures 13 and 14; (a) the aromatic ring/thick ring substituted hydrogen structure with wave numbers from 700 to 900 cm$^{-1}$, and the number of peaks is about seven to eight; (b) the oxygen-containing group and carbon skeleton structure with wave numbers from 1000 to 1780 cm$^{-1}$, and the number of peaks is about 14; (c) 2800–3000 cm$^{-1}$ for aliphatic groups, with four peaks; and (d) 3000–3680 cm$^{-1}$ for hydroxyl structures, with 6sixpeaks. The fitted peaks are all Gaussian peaks, and the fit is not less than 99.5%. The vibration forms of chemical bonds or functional groups corresponding to different wave numbers of FTIR are shown in Table 5 [28].

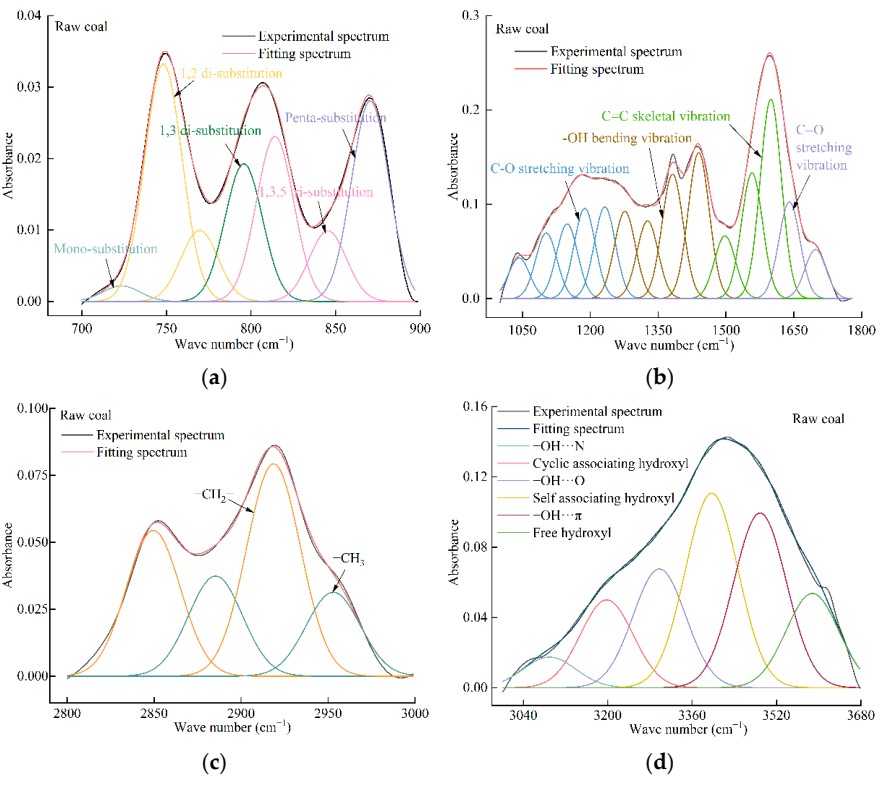

**Figure 13.** Peak fitting and peak assignment of RC. (**a**) The aromatic ring/thick ring-substituted hydrogen structure; (**b**) indicates the oxygen-containing group and carbon skeleton structure; (**c**) is for aliphatic groups; (**d**) indicates hydroxyl structures.

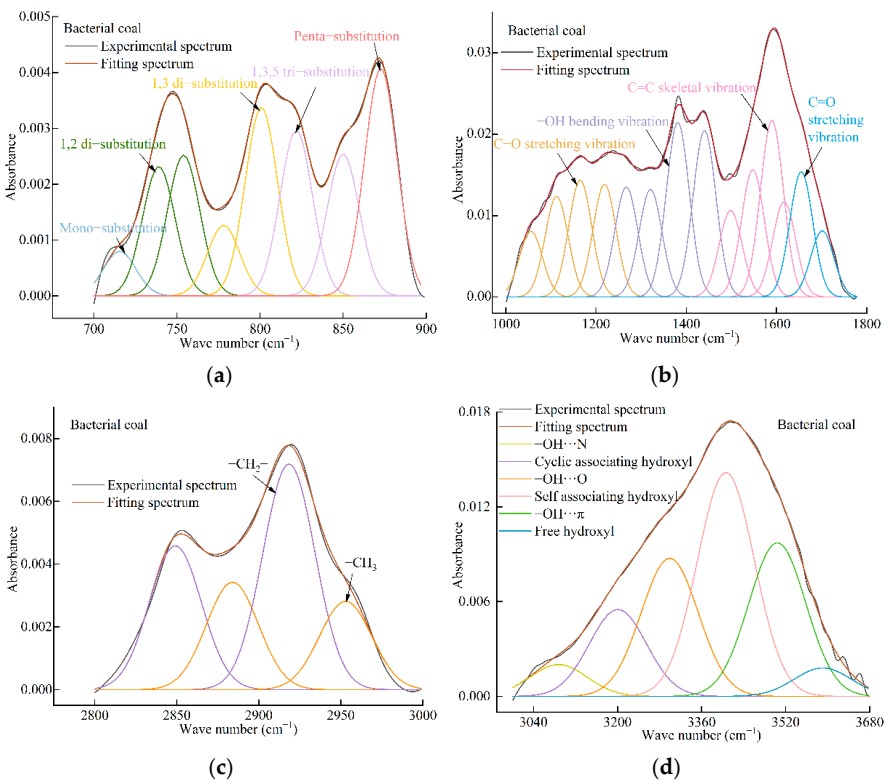

**Figure 14.** Peak fitting and peak assignment of BC. (**a**) Aromatic ring/thick ring-substituted hydrogen structure; (**b**) indicates the oxygen-containing group and carbon skeleton structure; (**c**) is for aliphatic groups; (**d**) indicates hydroxyl structures.

Infrared spectra were analyzed semi-quantitatively using the Lambert–Beer law based on the following equation:

$$A(v) = \lg \frac{1}{T(v)} = K(v)bc \tag{1}$$

In the equation:

$A(v)$—Absorbance of the sample spectrum at wave number $v$;

$T(v)$—The transmittance of the sample spectrum at wave number $v$;

$K(v)$—Absorbance coefficient of the sample at wave number $v$;

$b$—Thickness of the sample, cm;

$c$—the concentration of the sample, mol/L.

Currently, the peak height and peak area of infrared spectra are analyzed according to the Lambert–Beer law. The traditional manual separation method is inefficient, time-consuming, error-prone, and makes it easy to miss the target peak position, which cannot meet the demand of analyzing functional groups in coal. Since the infrared spectra are affected by the instrument and temperature, the peak area, which is less affected by the testing factors, is chosen as the analyzing factor to analyze the active groups.

From RC to BC, the percentage of mono- and meta-substitution in the aromatic structure increases and the total percentage of di-substitution decreases, where the percentage of o- substitution decreases. The slight increase in the percentage of 1,3,5 tri-substitution and the decrease in the penta-substitution indicate that this bacterium leads to a change in the aromatic nucleus substitution form, possibly due to cyclization of the aliphatic chain, dehydrogenation of the aromatic thick ring, or a change in the aromatic ring substitution position.

**Table 5.** FTIR wave number and corresponding chemical bond attribution.

| Wave Number (cm⁻¹) | Chemical Bond or Functional Group | Peak Area Percentage (%) | | Wave Number (cm⁻¹) | Chemical BOND or Functional Group | Peak Area Percentage (%) | |
|---|---|---|---|---|---|---|---|
| | | RC | BC | | | RC | BC |
| 700~730 | Mono-substitution | 1.7 | 4.0 | 2800~2850 | Methylene symmetric stretching vibration | 26.9 | 25.4 |
| 730~770 | Ortho-substitution | 34.3 | 24.4 | 2850~2890 | Methyl symmetric expansion vibration | 18.5 | 19.0 |
| 770~810 | Meta-substitution | 15.3 | 23.4 | 2900~2950 | Methylene anti-symmetric stretching vibration | 39.2 | 39.9 |
| 810~860 | 1,3,5 Tri-substitution | 26.2 | 27.7 | 2950~3000 | Methyl anti-symmetric telescopic vibration | 15.5 | 15.6 |
| 860~900 | Penta-substitution | 22.3 | 20.5 | 3000~3100 | Hydrogen bonds formed by hydroxyl and N atoms | 4.4 | 4.8 |
| 1000~1250 | C-O stretching vibration | 27.2 | 24.2 | 3100~3200 | Hydrogen bonds formed by cyclic tightly bonded hydroxyl groups | 12.5 | 13.1 |
| 1260~1350 | In-plane bending vibration in primary alcohol-OH | 12.4 | 13.3 | 3200~3300 | Hydrogen bond formed by hydroxyl group and ether oxygen | 17.0 | 20.8 |
| 1350~1410 | In-plane bending vibration in tertiary alcohol-OH | 9.4 | 10.7 | 3300~3400 | Hydrogen bonds formed by self-associative hydroxyl groups | 27.8 | 33.8 |
| 1410~1450 | Aromatic acid bending vibration in the hydroxyl plane | 11.0 | 10.2 | 3400~3500 | Hydrogen bonds formed by hydroxyl groups and π-bonds | 24.9 | 23.2 |
| 1450~1620 | Thick ring C=C skeleton expansion vibration | 29.1 | 29.8 | 3500~3600 | Free hydroxyl group | 13.4 | 4.3 |
| 1650~1740 | Aliphatic acid C=O stretching vibration | 11.0 | 11.7 | | | | |

The stretching vibration of C-O decreased by 3%, there was a small increase in the bending vibration in the hydroxyl plane, the percentage of hydroxyl groups of primary and tertiary alcohols increased little, and the C=O unsaturated bonds increased slightly. It is suggested that the C-O bond may be changed into an unsaturated bond by this bacterium in small amounts, however, the effect on the structure of C=C thick ring skeleton is not significant, so it is presumed that this bacterium has little ability to decompose the main structure of coal molecules, and has a small effect on oxygen-containing groups, which may change the percentage of coal side chain or heteroatom composition.

There was a small decrease in the methylene percentage and a small increase in the methyl percentage, which indicates that the involvement of the bacteriophage may break the break-prone bridge bonds in the coal structure to be formed into terminal groups, thus evolving towards stable molecular units.

The broad peaks from 3000 to 3680 cm$^{-1}$ are hydrogen bonds consisting of hydroxyl groups and $\pi$ bonds, as well as different peak overlaps such as coal-adsorbed water hydroxyl groups. Although the potassium bromide was pretreated by vacuum drying, the spectral peaks of adsorbed water hydroxyl groups were present in this band because of the easily absorbable water of potassium bromide and the experiments could not avoid the adsorption of water from the air [29]. The hydroxyl group showed the greatest degree of variation, with a 9.16% decrease in the percentage of free hydroxyl groups. Some scholars believe that the free hydroxyl vibration is mostly crystalline water of clay minerals, indicating that this bacterium may utilize the crystalline water in coal [30]. The percentage of hydrogen bonds formed by hydroxyl groups and $\pi$ bonds decreased by 1.75%, which may be due to the bacteria solution affecting some of the aromatic rings and reducing the density of $\pi$ electron clouds. The percentage of hydrogen bonds formed by self-associated hydroxyl groups increased by 6.04%, which may be due to the small amount of methane precipitated by the bacterial solution or the close spatial arrangement of the coal molecules, causing the percentage of self-associated hydrogen bonds to increase. The percentage of hydrogen bonds formed by hydroxyl ether oxygen increased by 3.89%, and the percentage of hydrogen bonds formed by both -OH---N hydrogen bonds and tightly connected cyclic -OH increased slightly.

*3.5. X-ray Photoelectron Spectral Analysis*

After X-ray irradiation, the intensity of photoelectrons emitted from the surface of the sample (I, referring to the peak area of the characteristic peak) has a linear relationship with the concentration of the atom (n) in the sample, and can, therefore, be utilized for semi-quantitative analysis of elements. This can be expressed simply as I = n*S, S being the sensitivity factor.

In Figure 15($a_1$–$c_1$), the XPS fitted spectra of C1s, O1s, and N1s in RC are shown, and (a2) (b2) (c2) indicate the XPS fitted spectra of C1s, O1s, and N1s in BC, respectively. Different electron binding energies correspond to different elements into peaks, and the elemental attribution forms are determined by the positions of the fitted peaks. Table 6 shows the electronic binding energies of BC and RC as well as the types and proportions of groups to which the peaks are attributed, including the characteristic groups of the elements carbon, oxygen, and nitrogen.

The carbon in coal is derived from organic matter, and the main types are carbon connected to aromatic structures and carbon connected to heteroatoms. It has four forms, as seen in Figure 15($a_1$,$a_2$). The electron-binding energy is from low to high, and the sub-peaks belong to aromatic rings and substituted alkanes, single-bonded phenolic or ether carbon (C-O), carbon double-bonded carbonyl form (C=O), and carboxyl form (COO-) in that order [31]. The C-C and C-H forms in RC and BC account for the largest proportion, at 32.15% and 37.03%, confirming that the main body of coal is a reticulated ring structure with carbon as the backbone, and the carboxyl carbon (COO-) form is the least, accounting for 18.33% and 6.30%. The number of single-bond carbon in RC is larger than that of double-bond carbon, and the ratio of their numbers is about 3:2. In BC, the ratio of phenolic

carbon or ether carbon quantity to carbonyl carbon quantity is about 1:1, which indicates that after the microbial action, the carbon and oxygen single bond is relatively reduced, while the carbon and oxygen double bond is relatively increased. This trend of change is consistent with the trend of FTIR spectra analysis results, and the percentage of carboxyl carbon decreases by 12.03%, which is the largest change. It indicates that this bacterium may consume carboxyl groups and the carboxyl carbon in coal may become the carbon source required for microbial life activities.

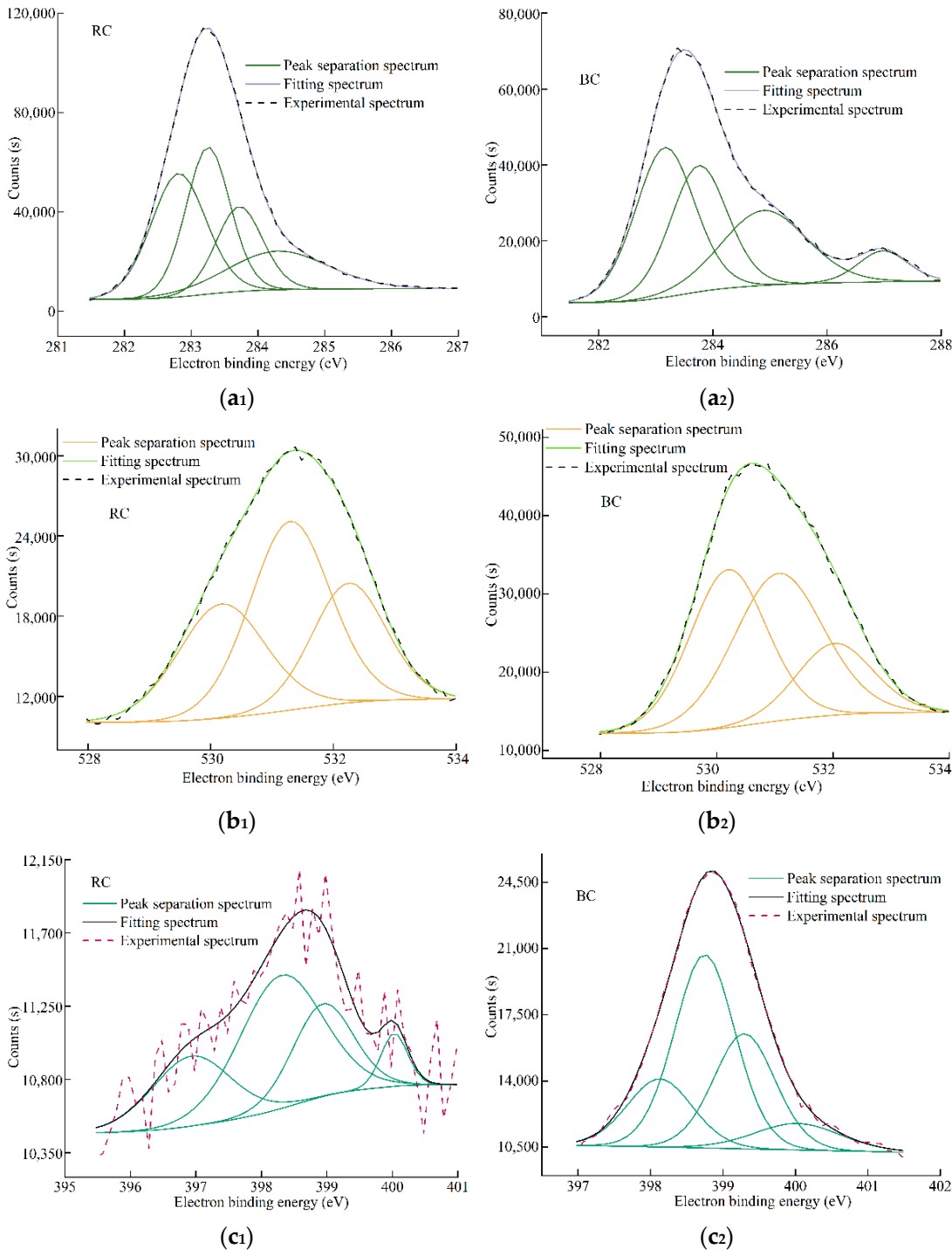

**Figure 15.** XPS segmented spectrum of elements C (1s), O (1s), and N (1s) of RC and BC. (**a₁**) C(1s) of RC; (**b₁**) O(1s) of RC; (**c₁**) N(1 s) of RC; (**a₂**) C(1s) of BC; (**b₂**) O(1s) of BC, and (**c₂**) N(1s) of BC.

**Table 6.** Electronic binding energy and proportion of elements C (1s), O (1s), and N (1s) of RC and BC.

| Elements | Characteristic Groups for Peak Attribution | (RC) Peak Position/eV | (BC) Peak Position/eV | (RC) Relative Content/% | (BC) Relative Content/% |
|---|---|---|---|---|---|
| C1s | Aromatic structures and alkane substitution groups | 282.81 | 283.14 | 32.15 | 37.03 |
| | Phenolic or ether carbon | 283.25 | 283.72 | 30.87 | 30.00 |
| | Carbonyl carbon (C=O) | 283.72 | 285.05 | 18.65 | 26.67 |
| | Carboxy carbon | 284.29 | 287 | 18.33 | 6.30 |
| O1s | Carbonyl oxygen (C=O) | 530.18 | 530.19 | 29.33 | 39.48 |
| | Carbon–oxygen single bond oxygen (C-O) | 531.29 | 531.06 | 44.44 | 42.92 |
| | Carboxyl oxygen | 532.25 | 532.02 | 26.22 | 17.59 |
| N1s | Nitrogen oxide (N-X) | 400.03 | 400.03 | 6.07 | 8.76 |
| | Protonated pyridine nitrogen (N-Q) | 398.94 | 399.3 | 23.83 | 27.65 |
| | Pyrrolizidine (N-5) | 398.3 | 398.75 | 46.73 | 46.08 |
| | Pyridine nitrogen (N-6) | 396.93 | 398.13 | 23.36 | 17.51 |

Organic oxygen in coal is dominated by carboxyl groups, carbon–oxygen unsaturated double-bonds (carbonyl), and carbon–oxygen saturated single-bonds (ether or phenol). There are three oxygen fitting peaks in Figure 15($b_1$,$b_2$) [32], and the fugitive forms of oxygen are oxygen in carbonyl form (C=O), carboxy oxygen single-bond oxygen in phenol or ether saturated form (C-O), and carboxyl oxygen (O=C-O) in order of electron-binding energy from low to high [33]. Of these, the largest amount of single-bonded oxygen is formed by ether or phenol. This may be due to the lone pair of electrons of oxygen in the phenol hydroxyl group and the conjugated structure formed by the thick ring, which is not easily broken. In RC, the ratio of the number of oxygen single bonds to double bonds is about 3:2. In BC, the ratio of the percent of single bonds to double bonds of oxygen is about 1:1, which is basically consistent with the conclusion of the C(1s) spectrum of XPS.

Nitrogen in coal is derived from vegetation, alkaloids, and bioproteins. The organic nitrogen fugacity is mainly in the form of pyridine nitrogen (N-6), pyrrole nitrogen (N-5), nitrogen oxide nitrate form nitrogen (N-X), and protonated pyridine nitrogen (N-Q) [34]. Figure 15($c_1$,$c_2$) represent the approximate fitted curves of nitrogen fugacity states, and the largest percentage of fitted peak area is usually for pyrrole-type nitrogen. Since plant chloroplasts contain a certain amount of nitrogenous metallocene structures during coal generation, the five-membered cyclic nitrogenous metallocene conjugation system is more stable. Therefore, the pyrrole nitrogen of RC and BC accounts for 46.73% and 46.08%, while the sum of N-6, N-Q, and N-X quantities account for 53.26% and 53.92% of the total nitrogen, indicating that the nitrogen fugitive in coal may be dominated by aryl fringe groups. However, as the bacterium metabolically multiplied in the coal, it affected the form of nitrogen present in the coal and reduced the percentage of pyridine nitrogen.

## 4. Conclusions

(1) The endogenous bacteria isolated from coal, named *Lysinibacilus* sp., is aerobic, with a complete growth cycle of about 70 h, and can grow in the coal environment and affect the surface morphology of coal, so that the surface of coal is smoothed and stomata develop;

(2) After the action of *Lysinibacilus* sp., the concentration of CO has significantly decreased, the oxidation characteristic temperature point in the low-temperature oxidation stage shifts back to different degrees than that of the raw coal. This bacterium could retard the desorption and precipitation of free water as well as physically adsorbed small molecule gases in the coal, and could delay the starting temperature of the thermal decomposition stage of the coal by 79.47 °C. Although this bacterium could not prevent coal ignition after coal combustion, *Lysinibacilus* sp. had a positive blocking effect on lignite during the

low-temperature oxidation stage of coal spontaneous combustion and reduced the total heat release of coal at this stage;

(3) *Lysinibacilus* sp. could affect the coal aromatic ring substitution mode. It changed the ratio of the percentage of C-O to C=O in lignite from 3:2 to about 1:1. This bacterium has weak capability to destroy the main part of coal structure, and has a certain effect on oxygen-containing groups and hydrocarbon groups. The hydroxyl group, carboxy carbon, and pyridine nitrogen in coal are affected by this bacterium to a greater extent, and it may use the hydroxyl group, carboxy carbon, and pyridine nitrogen in coal as a nutrient source to provide carbon and nitrogen sources required for reproductive metabolism.

**Author Contributions:** All authors contributed equally to this work. All authors have read and agreed to the published version of the manuscript.

**Funding:** This work was supported by the National Natural Science Foundation of China (grant numbers: 51974128).

**Institutional Review Board Statement:** Not applicable.

**Informed Consent Statement:** Not applicable.

**Data Availability Statement:** Data unavailability due to privacy.

**Conflicts of Interest:** The authors declare that they have no known competing financial interest or personal relationships that could have appeared to influence the work reported in this paper.

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
