# Peer review of "Study on the Influence of Coal Structure and Oxidation Performance by Endogenous Bacterium"

_fire, doi:10.3390/fire6090339_

Round 1
Reviewer 1 Report
The research method of this paper is reasonable and innovative. I maintain a positive attitude towards this paper, however, there are some minor shortcomings that need to be improved.
1. In section 3.1 of the article, what is the role of characteristic temperature point T3 and why is it missing in the table.
2. Although bacteria come from coal mines, it does not prove that they are safe. How can the author ensure that bacteria are non-toxic and harmless to the human body?
3. In the paper, the temperature range for DSC analysis of coal samples to calculate the heat release is chosen to be 28-800 °C. Why not choose the temperature range that is generally considered to be the self-heating stage? For example, room temperature -90 °C. The authors are requested to answer this question.
4. On approximately the third page, in line 144, the three forms of expression ‘bacteria X’, ‘X bacterium’ and ‘bacterium X’ appear, and I suggest that it would be better to harmonize them.
Author Response
Dear reviewer:
I am very grateful to reviewer 1 for recognizing my experiment and thesis in his hectic schedule. This will have a positive effect on my academic career, allowing me to be encouraged and giving me more confidence to face future studies. In view of the suggestions given by experts, my reply is revised as follows.
Suggestion 1. In section 3.1 of the article, what is the role of characteristic temperature point T3 and why is it missing in the table.
Reply the suggestion:After the temperature point T2, the oxygen consumption and gas desorption reach a dynamic equilibrium, resulting in the quality of the coal sample remaining stable for a period of time. The active temperature point T3 is the temperature point at which the mass of the coal sample remains constant until the start of weight gain. The oxidization and weight gain stage starts from T3, when the dynamic equilibrium of coal oxygen adsorption and desorption is broken. A large number of groups in the coal molecular structure are activated and oxidization reaction starts, the chemical adsorption increases rapidly, the adsorption amount is larger than the gas escape amount, and the coal samples start to gain weight gradually.
The coal sample used in the experiment was lignite, which has a low degree of metamorphism and is susceptible to thermal decomposition during heating. Thus, a large amount of gas escapes, so that the mass loss is always greater than the amount of coal oxygen adsorption weight gain. There is no stage in the TG curve where the coal mass remains constant, so no T3 temperature point exists.
Suggestion 2. Although bacteria come from coal mines, it does not prove that they are safe. How can the author ensure that bacteria are non-toxic and harmless to the human body?
Thanks to the experts who asked questions about the safety of bacteria, human beings are still an indispensable labor force in the production of coal mines, so the issue of safety should not be ignored. There are many types of bacteria in coal mines, and we have isolated several bacteria from the coal mine environment, in order to ensure their safety, it is necessary to test the species of bacteria in order to determine whether they are pathogenic or not, so in this paper, we screened the non-pathogenic bacteria as a raw material for the inhibition of coal. Therefore, the source of the bacteria does not guarantee its safety, however, the safety can be determined by testing the genetic sequence of the bacteria.
Suggestion 3. In the paper, the temperature range for DSC analysis of coal samples to calculate the heat release is chosen to be 28-800 °C. Why not choose the temperature range that is generally considered to be the self-heating stage? For example, room temperature -90 °C. The authors are requested to answer this question.
The oxidation process of coal is exothermic, and the macroscopic exotherm measured by the DSC curve is the combined result of the heat of evaporation of moisture, the release of heat from volatile products, the exotherm of oxidation reaction, and the thermal effect of coal body heating. 28 to 800°C would include the entire phase of the coal from evaporation of moisture to combustion during the increase in temperature. Part of the heating stage of 28 ~ 800℃ is not enough to represent the whole process of coal oxygen reaction. The area surrounded by DSC curve is the total heat release of the whole heating, and the main oxidation heat release peak is formed by the main components of the coal sample, so only the area of room temperature ~ 90℃ could not represent the total heat release of the whole reaction process.
Suggestion 4. On approximately the third page, in line 144, the three forms of expression ‘bacteria X’, ‘X bacterium’ and ‘bacterium X’ appear, and I suggest that it would be better to harmonize them.
Thanks for the careful review by experts. I have corrected them in the article, changed them ‘bacterium X’ and highlighted them in the manuscript.

Reviewer 2 Report
Dear authors,
I think this manuscript is well organized, logical, and structured. However, there are several issues with this manuscript that need to be addressed before publication.
(1) Page 2, line 51, I don't think "He proposed" is appropriate here.
(2) page 6, I would suggest that Table 2 should be added the content of elemental sulfur.
(3) Page 8, the description of Figure 10 in the main text is, in my opinion, unclear, and I suggest that it be improved.
(4) Page 9, I think the title of Figure 10 is inaccurate. It should not be the CO concentration of the two coals, but the CO concentration released by the heat of the two coals.
(5) Page 13, could Table 5 be removed from the manuscript?
(6) The results in Table 6 are not described in the main text.
Author Response
Dear reviewer:
Thank you for your decision and constructive comments on my manuscript. We have carefully considered the suggestion of reviewer and make some changes.
(1) Page 2, line 51, I don't think "He proposed" is appropriate here.
Thanks to the experts for their special attention to the manuscript in details that will help me a lot. The page 2, line 51, has been modified to read ' the judgment indicators of ignition degree in closed region are proposed', and highlight it in the manuscript.
(2) page 6, I would suggest that Table 2 should be added the content of elemental sulfur.
Supplemental sulfur reflects the integrity of the coal element. I have added the total sulfur content in Table 2 and highlighted it in the manuscript.
(3) Page 8, the description of Figure 10 in the main text is, in my opinion, unclear, and I suggest that it be improved.
On page 8, I have corrected the textual description of Figure 10, which has been highlighted in the text. ‘During the initial oxidation stage (60~80°C), there was no significant difference in the CO concentration generated by heating between BC and RC. However, during the slow self-heating stage (80~140℃), the CO concentration generated by coal after bacterial reaction differs from that of raw coal, with the CO concentration of BC slightly lower than that of RC. The critical acceleration stage (140~180°C) is the stage with the largest difference in CO concentration, which of BC is significantly smaller than the RC at the same temperature. During the pyrolysis fission stage (>180°C), the CO concentration of BC remains consistently less than RC. CO was used as an indicator gas to determine the degree of coal oxidation and it was found that the coal treated with Lysinibacillus sp. has a diminished ability to participate in reactions with oxygen.’ has been added in the manuscript.
(4) Page 9, I think the title of Figure 10 is inaccurate. It should not be the CO concentration of the two coals, but the CO concentration released by the heat of the two coals.
As for the suggestion made by improper expression in the manuscript, I changed the picture title of Figure 10 to ‘CO concentration released by RC and BC during heating’.
(5) Page 13, could Table 5 be removed from the manuscript?
Figure 5 in the manuscript shows the attribution and proportions of coal at different band peaks. In the papers of other researchers, table 4 of the article ‘Microcharacteristic analysis of CH4 emissions under different conditions during coal spontaneous combustion with high-temperature oxidation and in situ FTIR’ by Zhao et al. has the types of IR peaks and their attribution (https://doi.org/10.1016/j.energy.2020.118494). Table 3 of the article ‘Use of FTIR, XPS, NMR to characterize oxidative effects of NaClO on coal molecular structures’ by Jing et al. has the type of infrared spectral peak, wave number, and peak area (https://doi.org/10.1016/j.coal.2018.11.017). Table 1 of the article ‘Mechanism of oxidation of low rank coal by nitric acid’ by Shi et al. has the absorption peak area and proportion of the main functional groups in infrared spectroscopy (DOI: 10.1007/s12404-012-0411-6). Therefore, I think Table 5 is somewhat necessary, and have not removed Table 5.
(6) The results in Table 6 are not described in the main text.
Thanks for the expert suggestion, it was my mistake, so I added the description of Table 6 to the manuscript. ‘Table 6 shows the electronic binding energies of BC and RC as well as the types and proportions of groups to which the peaks are attributed, includes the characteristic groups of the elements carbon, oxygen and nitrogen.’

Reviewer 3 Report
The article discusses using aerobic endogenous bacteria isolated from coal to prevent spontaneous combustion in enclosed mining areas. The bacteria, specifically Lysinibacillus sp., were found to delay coal oxidation and reduce heat release during combustion. Structural analyses confirmed chemical changes in coal due to bacterial action. The study suggests the potential of these microorganisms for enhancing mining safety by mitigating coal spontaneous combustion.
The article is interesting, and the conducted research is substantiated. The visual presentation is well-executed. The research results are presented clearly. Microscopic images are accurately selected. Nevertheless, certain revisions are necessary before the article can be published:
1. Table 2 presents the elemental analysis of coals. It would be beneficial for the study to include sulfur content;
2. The title of Figure 6 should include information about the imaging mode used (SE or BSE);
3. In the description of chapter 2.2, imaging parameters and the method of conducting elemental analysis should be added;
4. Line 241: redundant space „heating -gas chromatography”;
5. It is interesting that only apart from the T7 temperature (Table 3), the temperatures for the BC sample are higher. References to studies by other authors and the confrontation of this observation should be supplemented here. In addition, reference should be made to what happens to the carbon structure at these temperatures and how bacteria affect the structure;
6. It would be advantageous if Figures 12a and 12b appeared first, followed by Table 4;
7. Chapter 3.3 should confront the results with TG-DTG tests;
8. Line 342: it should be „The integration…”; Line 346: it should be „The DSC…”;
9. Chapter 3.4 lacks a description of how the series of intermediate signals in Figures 13 and 14 were extracted. What models were used? It is necessary to provide a description of this procedure;
10. In Table 5, the formatting of rows in the column "Chemical bond..." should be standardized – whether using uppercase or lowercase letters;
11. Chapter 3.5 lacks a description of how the series of intermediate signals in Figure 15 were extracted. It is necessary to provide a description of this procedure;
12. It would be more advantageous if Figure 16 appeared before Table 6.
Author Response
Dear reviewer:
On behalf of my co-authors, we thank you very much for giving us an opportunity to revise our manuscript. Your comments and suggestions are valuable and helpful for our future research, we have carefully studied all the comments and corrected them carefully, the revisions are highlighted in the manuscript, the response to the reviewers is as follows.
- Table 2 presents the elemental analysis of coals. It would be beneficial for the study to include sulfur content;
I thank the reviewers for the thoughtful suggestions on details. I have added the total sulfur content in Table 2 and highlighted it in the manuscript.
- The title of Figure 6 should include information about the imaging mode used (SE or BSE);
I have supplemented its imaging mode in the manuscript and highlighted it. ‘Figure 6. Scanning electron micrographs of secondary electron (SE) imaging modes of RC and BC.’
- In the description of chapter 2.2, imaging parameters and the method of conducting elemental analysis should be added;
In Section 2.2, I emphasize and add what is missing from the manuscript as follows. ‘the sample structure was observed using secondary electron imaging mode (SE).’ ‘Ultimate analysis was carried out with the Germany Elementar-UNICUBE, with a sample weight of about 0.20 mg. Burn the sample at high temperature until it decomposes, and the elements to be measured were converted into the corresponding gases for analysis.’
- Line 241: redundant space „heating -gas chromatography”;
In the picture title of Figure 7, there appears ‘heating-gas chromatography’, and I have removed this space and highlighted it.
- It is interesting that only apart from the T7 temperature (Table 3), the temperatures for the BC sample are higher. References to studies by other authors and the confrontation of this observation should be supplemented here. In addition, reference should be made to what happens to the carbon structure at these temperatures and how bacteria affect the structure;
This contributes to the completeness of the thesis statement. However, I feel very regretful that due to the lack of references related to keywords such as bacteria, coal, and combustion, I have cited a reference here that relates to the confrontation between the observation in my manuscript, which explains that after combustion bacterial residue shifts the temperature point of coal forward. The supplementary content has been highlighted. ‘The conclusion of the maximum combustion rate T7 after the coal has been burned is in agreement with Hong et al. The maximum combustion rate of the bacterial residue corresponds to a significantly lower temperature than that of the coal [26].’
Through experimental analysis, the bacterium may cause a decrease in the relative amount of carbon-oxygen single bonds and an increase in carbon-oxygen double bonds, while the proportion of carboxyl carbon decreases and the carboxyl group changes to the greatest extent. The carboxyl structures in the coal may be participating in the growth of the bacterium or reacting with the metabolites of the bacterium. In response to the suggestions provided by the reviewers, in future research, we will focus on studying the mechanism of action between bacteria and coal, as well as the complex chemical changes involved. Thanks to the Reviewers. I believe that this revised paper has been improved considerably.
- It would be advantageous if Figures 12a and 12b appeared first, followed by Table 4;
The order of the figures and tables noted by the reviewer is important, and I have swapped the positions of Figure 12(a)(b) and Table 4.
- Chapter 3.3 should confront the results with TG-DTG tests;
We thank the reviewers for asking questions about the content of the manuscript, the connection that should be made between the two is something neglected. I have added and refined the link between TG-DTG and DSC in the manuscript. The modified traces are highlighted. ‘In the water evaporation desorption stage. TS0 indicates the maximum heat absorption temperature, and BC is delayed by 20.86°C compared with RC. Tf0 indicates the end temperature of heat absorption by water evaporation, which is close to the T2 temperature point on the TG curve. RC is 117.88°C and BC is 129.83°C, with a difference of 11.95°C. At the combustion stage, TS1 indicates the maximum peak temperature of DSC in the exothermic phase, which has all higher temperature point than T7. This is due to the increase in temperature caused by heating, which reduces the combustible components of the coal and continually decomposes to produce a large number of gaseous products, thereby driving the enthalpy to its peak. In addition, TS1 of BC is 32.02°C higher than the temperature point of RC.’
- Line 342: it should be „The integration…”; Line 346: it should be „The DSC…”;
I should learn from the seriousness and rigor of the reviewer. Please forgive my carelessness, which I have corrected in the manuscript.
- Chapter 3.4 lacks a description of how the series of intermediate signals in Figures 13 and 14 were extracted. What models were used? It is necessary to provide a description of this procedure;
The range of wave numbers detected by conventional infrared spectroscopy, i.e., wave numbers 400-4000 cm-1, corresponds exactly to the wavelength region of the mid-infrared. In a compound molecule, the atoms that make up the chemical bond or functional group are in a state of constant vibration at a frequency comparable to the vibrational frequency of infrared ray. Therefore, when a compound molecule is irradiated with infrared ray, the chemical bonds or functional groups in the molecule can selectively absorb certain wavelengths of infrared rays, which causes the transition of rotational and vibrational energy levels in the molecule. The infrared absorption spectrum of the substance can be obtained by detecting the absorption of these infrared rays. Due to the different absorption frequencies of different chemical bonds or functional groups, the positions on the infrared spectrum are not the same, while the majority of organic and inorganic substances appear in the mid-infrared region of the fundamental frequency absorption band. Figures 13 and 14 show the spectra of Fourier transform infrared (FTIR) spectra with split-peak fitting. I have added the principle of infrared spectral peak-splitting signal in the manuscript. ‘Infrared spectra were analyzed semi-quantitatively using the Lambert-Beer law based on the following equation:
(1)
In the equation:
A(v) - absorbance of the sample spectrum at wave number v;
T(v) - The transmittance of the sample spectrum at wave number v;
K(v) - absorbance coefficient of the sample at wave number v;
b - thickness of the sample, cm;
c - is the concentration of the sample, mol/L.
Currently, the peak height and peak area of infrared spectra are analyzed according to the Lambert-Beer law. The traditional manual separation method is inefficient, time-consuming, error-prone and easy to miss the target peak position, which cannot meet the demand of analyzing functional groups in coal. Since the infrared spectra are affected by the instrument and temperature, the peak area, which is less affected by the testing factors, was chosen as the analyzing factor to analyze the active groups.’
- In Table 5, the formatting of rows in the column "Chemical bond..." should be standardized – whether using uppercase or lowercase letters;
Forgive me again for my carelessness. I have standardized the format of the chemical bond column in Table 5.
- Chapter 3.5 lacks a description of how the series of intermediate signals in Figure 15 were extracted. It is necessary to provide a description of this procedure;
The figure shows the principle of X-ray photoelectron spectroscopy signal generation, which belongs to the surface analysis means, the fundamental reason is that although X-rays can penetrate the sample to a great depth, but only a thin layer of the sample near the surface of the emitted photoelectrons can escape. The depth of detection (d) of the sample is determined by the depth of escape of the electrons (λ, which is affected by the wavelength of the X-rays and the state of the sample, etc.). When a beam of photons irradiated to the surface of the sample, the photons can be absorbed by the electrons in the atomic orbitals of an element in the sample, causing the electrons to break away from the atomic nucleus, and to be emitted with a certain amount of kinetic energy from the interior of the atom into free photoelectrons, while the atom itself is turned into an excited state of the ion.
According to Einstein's law of photoelectric emission there is:
Ek =hν- EB
In the equation:
Ek is the kinetic energy of the emitted photoelectron;
hν is the energy of X-ray source photons;
EB is the binding energy on a specific atomic orbital (different atomic orbitals have different binding energies).
It can be seen from Eq. The energy of photoelectrons is characteristic for a particular monochromatic excitation source and a particular atomic orbital. When the energy of the excitation source is fixed, the energy of its photoelectrons is only related to the species of the element and the atomic orbitals excited by ionization. Therefore, we can qualitatively analyze the elemental species of a substance based on the binding energy of the photoelectrons.
It was added in the manuscript. ‘After X-ray irradiation, the intensity of photoelectrons emitted from the surface of the sample (I, referring to the peak area of the characteristic peak) has a linear relationship with the concentration of the atom (n) in the sample, and can therefore be utilized for semi-quantitative analysis of elements. This can be expressed simply as I = n*S, S being the sensitivity factor.’
For a solid sample of two elements i and j, if they are known to the sensitivity factor Si and Sj, and then measured the intensity of their specific spectral lines Ii and Ij, then the ratio of their atomic concentration: ni: nj = (Ii/Si): (Ij/Sj), so you can get the relative content of the elements.
- It would be more advantageous if Figure 16 appeared before Table 6.
I have swapped the position of the figure with Table 6, aligning the figure before the table.
